# Divergent Evolution of Eukaryotic CC- and A-Adding Enzymes

**DOI:** 10.3390/ijms21020462

**Published:** 2020-01-10

**Authors:** Lieselotte Erber, Paul Franz, Heike Betat, Sonja Prohaska, Mario Mörl

**Affiliations:** 1Institute for Biochemistry, Leipzig University, Brüderstraße 34, 04103 Leipzig, Germany; lieselotte.erber@uni-leipzig.de (L.E.); paul.franz@genetik.uni-halle.de (P.F.); heike.betat@uni-leipzig.de (H.B.); 2Computational EvoDevo Group, Department of Computer Science, Leipzig University, Härtelstraße 16-18, 04107 Leipzig, Germany; sonja@bioinf.uni-leipzig.de; 3Interdisciplinary Center for Bioinformatics, Leipzig University, Härtelstraße 16-18, 04107 Leipzig, Germany; 4Santa Fe Institute for Complex Systems, 1399 Hyde Park Road, Santa Fe, NM 87501, USA

**Keywords:** tRNA nucleotidyltransferase, enzyme evolution, *Salpingoeca rosetta*, *Schizosaccharomyces pombe*

## Abstract

Synthesis of the CCA end of essential tRNAs is performed either by CCA-adding enzymes or as a collaboration between enzymes restricted to CC- and A-incorporation. While the occurrence of such tRNA nucleotidyltransferases with partial activities seemed to be restricted to Bacteria, the first example of such split CCA-adding activities was reported in *Schizosaccharomyces pombe*. Here, we demonstrate that the choanoflagellate *Salpingoeca rosetta* also carries CC- and A-adding enzymes. However, these enzymes have distinct evolutionary origins. Furthermore, the restricted activity of the eukaryotic CC-adding enzymes has evolved in a different way compared to their bacterial counterparts. Yet, the molecular basis is very similar, as highly conserved positions within a catalytically important flexible loop region are missing in the CC-adding enzymes. For both the CC-adding enzymes from *S. rosetta* as well as *S. pombe*, introduction of the loop elements from closely related enzymes with full activity was able to restore CCA-addition, corroborating the significance of this loop in the evolution of bacterial as well as eukaryotic tRNA nucleotidyltransferases. Our data demonstrate that partial CC- and A-adding activities in Bacteria and Eukaryotes are based on the same mechanistic principles but, surprisingly, originate from different evolutionary events.

## 1. Introduction

In all domains of life, tRNAs play a crucial role during translation. At their 3′ terminus, these molecules carry the invariant sequence cytidine–cytidine–adenosine (CCA), which is important for the attachment of the corresponding amino acid and for proper participation of the tRNA in translation [1,2,3]. In all Eukaryotes, most Archaea, and many Bacteria, the CCA triplet is not encoded in the tRNA genes [4,5,6,7,8] and has to be added post-transcriptionally by ATP(CTP):tRNA nucleotidyltransferases (CCA-adding enzymes). These enzymes belong to a family of specialized polymerases with a unique mechanism of polymerization without the need of a nucleic acid-based template [3,9,10,11]. In organisms that encode the CCA triplet in their tRNA genes, the CCA-adding enzyme is involved in repair and maintenance of the cellular tRNA pool [12,13]. Intriguingly, CCA-adding enzymes also play a role in different layers of tRNA quality control to identify damaged and nonfunctional tRNAs [14,15,16].

Based on conserved sequence patterns in the catalytic core, tRNA nucleotidyltransferases are recognized as members of the polymerase β superfamily [17,18] and can be divided into class I (archaeal) and class II (bacterial and eukaryotic), depending on structural similarities and their mode of action [11,19,20,21]. In class II CCA-adding enzymes, a set of highly conserved motifs in the N-terminal catalytic core is responsible for proper CCA-addition [10,11,22] (Figure 1A). In the single nucleotide-binding pocket (motif D; EDxxR), conserved amino acids serve as a polymerization template, forming Watson–Crick-like hydrogen bonds with the incoming nucleotides [10]. The specificity switch from CTP to ATP binding is mediated by a flexible loop acting as a lever to adjust the relative orientation of the templating side chains for interaction with either CTP or ATP [22,23,24,25,26]. Deletions, as well as most point mutations in this region, have a strong inhibitory effect on the incorporation of the terminal A residue, underlining its importance for a complete CCA synthesis. For addition of the terminal A, a conserved region consisting of basic (B) and acidic (A) residues (B/A motif) upstream of the flexible loop is involved in proper positioning of the 3′OH for nucleophilic attack on the incoming ATP [11,22,27].

For an increasing number of bacteria, it is reported that CCA-addition is performed by the collaborative action of enzymes with partial activities [22,28,29,30,31,32]. While both types of enzymes share the five core motifs with class II CCA-adding enzymes, the flexible loop region is absent in CC-adding enzymes, inhibiting the switch from CTP to ATP specificity. This deletion suggests that bacterial CC-adding enzymes originated from CCA-adding enzymes [22,33]. In A-adding enzymes, on the other hand, the structure of the C-terminal part (located outside of the catalytic core) is equipped with a physical constraint that prevents short tRNAs without a 3′ CC end from entering the catalytic core for A-addition [32,34].

Recently, we showed in an extended phylogenetic analysis that metazoan CCA-adding enzymes originate from a horizontal gene transfer event (HGT) from Alphaproteobacteria to the common ancestor of Metazoa and their unicellular relatives, the Choanozoa [8]. In the course of evolution, the newly acquired alphaproteobacterial gene replaced the original ancestral eukaryotic gene (Figure 1B). Plants, Fungi, and Amoebozoa, however, carry the ancestral eukaryotic CCA-adding enzyme (Figure 1B). Consequently, eukaryotic tRNA nucleotidyltransferases can be subdivided into acquired alphaproteobacterial-like and ancestral eukaryotic CCA-adding enzymes (a-type enzymes, e-type enzymes), both showing characteristic differences in their catalytic core motifs [8]. Surprisingly, in the choanoflagellates *Salpingoeca rosetta* and *Monosiga brevicollis*, both an a-type as well as an e-type enzyme are encoded in their genomes. In these organisms, the original eukaryotic gene was not replaced and still exists in parallel with the alphaproteobacterial-like version [8]. In addition, representatives of the group of Fungi (e.g., *Schizosaccharomyces pombe*) also possess two versions of nucleotidyltransferases [35,36,37,38]. However, they are both of ancestral eukaryotic (e-type) origin [8].

Here, we present an in-depth phylogenetic analysis indicating that the occurrence of multiple nucleotidyltransferases is not limited to Bacteria, but is equally found in Eukaryotes. Similar to the situation in *S. pombe* [37,38], the enzymes of *S. rosetta* (group of Choanoflagellata) exhibit separate activities for CC- and A-addition and have to collaborate for complete CCA synthesis. A detailed bioinformatical analysis allowed us to pin down the unusual evolution of eukaryotic CC- and A-adding enzymes in detail. Surprisingly, the eukaryotic CC-adding enzymes do not carry the deletion of the flexible loop element, a hallmark for the bacterial counterparts [22]. Yet, our analyses also show that in the eukaryotic CC-adding enzymes, this region is affected by an accumulation of point mutations responsible for the loss of the A-adding activity. Accordingly, the insertion of a loop sequence of a CCA-adding enzyme closely related to the CC-adding enzyme of Choanoflagellata (in the case of *S. rosetta*; Figure 1C) or Taphrinomycotina (in the case of *S. pombe*; Figure 1D) restores A-adding activity and converts the CC-adding enzymes into CCA-adding versions in both cases.

## 2. Results

### 2.1. The Occurrence of More Than One tRNA Nucleotidyltransferase Is Also Common in Eukaryotes

Phylogenetic analyses by us and others indicated that the occurrence of multiple sequences of candidate tRNA nucleotidyltransferases might be more common among genomes of major eukaryotic taxa—i.e., Holozoa, Fungi, Algae, Amoebozoa, and Plants—than previously anticipated [8,35] (Figure 1B). In order to ascertain that the presence of multiple genes is not the result of incorrect genome assembly or annotation errors, we refined our first analysis and focused on the computational and functional characterization of duplicate tRNA nucleotidyltransferases in the group of Choanoflagellata and *Schizosaccharomyces*.

In the group of Choanoflagellata, the two open reading frames for tRNA nucleotidyltransferases in *S. rosetta* and *M. brevicollis* [8] prompted us to examine transcriptome data of 19 additional Choanoflagellata [39]. Using the *S. rosetta* sequences as a query in tblastn searches, we found candidate sequences for a-type tRNA nucleotidyltransferases in all choanoflagellate transcriptomes, while candidate sequences for e-type enzymes could be retrieved only in nine cases (Figure 1C; Appendix A). The bioinformatic classification as a-type or e-type enzymes based on sequence similarity and common motifs [8] is further supported by the phylogenetic network shown in Figure 2. Here, the prominent split clearly separates the sequences into two groups, consistent with the classification into a-type and e-type enzymes of different evolutionary origin [8].

The examination of the genomic sequence of *Schizosaccharomyces pombe* revealed two sequences with high similarity (>32% at protein level) to the known CCA-adding enzyme of *Saccharomyces cerevisiae* [8,37,38]. To address whether this observation is restricted to *S. pombe* or whether it is more widespread among Fungi, we performed database searches and phylogenetic analyses on all available genomes of the genus *Schizosaccharomyces* as well as closely related species. The results indicate that all four *Schizosaccharomyces* species (*S. pombe*, *S. japonicus*, *S. cryophilus*, and *S. octosporus*) encode two candidate tRNA nucleotidyltransferases, while all other members of the group of Taphrinomycotina show genomic evidence for only one CCA-like enzyme (Figure 1D). In the phylogenetic network depicted in Figure 3, both candidate tRNA nucleotidyltransferases of the four *Schizosaccharomyces* species cluster with the e-type enzymes, represented by Fungi and Plants. A prominent split separates these e-type enzymes from the phylogenetically diverse set of a-type enzymes, including all variants of bacterial tRNA nucleotidyltransferases (CCA-, CC-, and A-adding enzymes) as well as a-type enzymes of Metazoa (Figure 3). The bacterial type IIa CCA-adding enzymes, closely related to the A-adding enzymes of this kingdom, are also found in this branch, corroborating our original phylogenetic identification of this type of enzyme [32].

Both enzyme copies in *S. rosetta* as well as in *S. pombe* show all five elements of the catalytic core [10] and do not carry a deletion of the flexible loop element, the hallmark for bacterial CC-adding enzymes (Figure 4A and Figure 5A) [11,22]. However, Reid et al. and Preston et al. demonstrated that the *S. pombe* enzymes exhibit partial activities and represent the first example of eukaryotic CC- and A-adding enzymes [37,38]. The CC-adding enzyme lacks the B/A motif that positions the 3′-end of the reaction intermediate tRNA-CC for the terminal A-addition [11,22,27,37]. Hence, its absence is an indication for the altered catalytic activity of this enzyme. In the choanoflagellates, however, both enzyme copies carry the B/A motif as well as the flexible loop element (Figure 1A), such that a reliable estimation of their catalytic activity is not possible. Furthermore, computer-based predictions indicate that the N-terminus of these enzymes likely represents a mitochondrial import sequence (https://bio.tools/MITOPROT_II) [40]. However, the accuracy of such predictions is usually only moderate, and the import probability differs from enzyme to enzyme. As the only available choanoflagellate mitochondrial genome of *Monosiga brevicollis* carries only tRNA genes lacking the CCA end, it is highly likely that at least one enzyme (with CCA-adding activity) or both (with split activities) have to be imported into the organelle. To shed some light on the actual activity of such a choanoflagellate enzyme pair, we investigated the individual properties of both enzymes of *S. rosetta*.

### 2.2. CCA-Addition in S. rosetta Requires Collaboration of CC- and A-Adding Enzymes of Different Evolutionary Origin

To determine the activity of the a-type and e-type enzymes of *S. rosetta*, the corresponding recombinant proteins were expressed in *Escherichia coli* and tested in vitro for activity. To this end, radioactively labeled yeast tRNA^Phe^, either lacking the complete CCA terminus (tRNA) or ending with two C residues (tRNA-CC), was incubated with the respective enzyme in the presence of ATP, CTP, or all four NTPs. Both *S. rosetta* tRNA nucleotidyltransferases were catalytically active, however, with opposing preferences for tRNA substrates and nucleotides (Figure 4B). The a-type enzyme exclusively adds two C residues on a tRNA lacking the CCA end while no nucleotide incorporation could be detected on a tRNA carrying the CC end (Figure 4B, left). By contrast, the e-type enzyme exhibits no activity on the tRNA substrate lacking the CCA end, whereas a single A residue is added to a tRNA-CC (Figure 4B, right). Furthermore, the combination of both enzymes efficiently synthesizes a complete CCA terminus on a tRNA substrate in vitro (Appendix A). Hence, *S. rosetta* performs CCA-addition in a collaboration of two tRNA nucleotidyltransferases with partial activities, similar to *S. pombe* and several bacterial species with separate activities for CCA-addition [22,28,29,30,31,32,37,38]. Yet, the two pairs of eukaryotic CC- and A-adding enzymes differ dramatically in their evolutionary origin. In *S. pombe*, both enzymes represent the ancestral eukaryotic type (e-type). In *S. rosetta*, however, only the A-adding enzyme is an e-type enzyme, while the CC-adding enzyme is of alphaproteobacterial origin (a-type).

### 2.3. Introduction of a Conserved Loop Region Restores CCA-Adding Activity of S. rosetta CC-Adding Enzyme

A sequence comparison of the *S. rosetta* a-type and e-type enzymes with eukaryotic a-type and e-type CCA-adding enzymes revealed a high conservation of the catalytic core motifs A to E and the basic/acidic motif (B/A) (Figure 4A). Similar to the *S. pombe* CC-adding enzyme [37] and in contrast to its bacterial counterparts [22], the *S. rosetta* a-type enzyme with CC-adding activity carries a flexible loop element without deletions. Compared to the related a-type enzymes of *Homo sapiens* and *Caenorhabditis elegans*, however, this region exhibits several strong sequence deviations in the N-terminal part. Only in the C-terminal part, several residues match the corresponding positions found in other eukaryotic a-type enzymes [25]. In bacterial CC-adding enzymes, this loop region is deleted, and the enzymes are locked in a conformation competent for CTP binding [22]. Hence, we hypothesized that the observed sequence differences are the cause for the restricted activity of the CC-adding enzyme of *S. rosetta*.

To investigate this possibility, we replaced the existing loop sequence of the *S. rosetta* CC-adding enzyme with the corresponding functional loop of the related human CCA-adding enzyme that differs at 13 positions, while eight (mostly C-terminal) residues are identical (Figure 4A,C). Intriguingly, the resulting enzyme chimera, SHS, is able to add a complete CCA triplet to the tRNA substrate and a terminal A residue to the substrate tRNA-CC (Figure 4D). Furthermore, in the presence of ATP only, the chimera adds a single A residue to a tRNA without 3’-CC, indicating a certain loss of specificity, as it is also described for some bona fide CCA-adding enzymes [23,41,42]. Nevertheless, the insertion of a conserved and functional loop from an a-type CCA-adding enzyme in the *S. rosetta* a-type CC-adding enzyme restores its CCA-adding activity.

### 2.4. Schizosaccharomyces pombe Has Two Ancestral Eukaryotic Nucleotidyltransferases with Separate Activities

The two *S. pombe* e-type enzymes were recently identified as the first eukaryotic pair of CC-and A-adding enzymes [37,38]. Interestingly, the *S. pombe* CC-adding enzyme also carries a loop sequence that differs at up to 14 residues compared to other e-type CCA-adding enzymes in the alignment, again predominantly located in the N-terminal part of the loop (Figure 5A). However, and in contrast to the *S. rosetta* enzyme, the tRNA 3′-end-binding B/A motif is also affected, and only the basic residue is present [25,27,37]. Restoration of this motif alone is not sufficient to restore CCA-addition in this enzyme, as recently demonstrated [37]. To investigate whether the sequence deviations in the loop element also contribute to the restricted activity of this enzyme (similar to the *S. rosetta* situation), we restored the B/A motif and replaced the mutated loop region of this enzyme with the corresponding element of the closely related fungal e-type CCA-adding enzyme of *Taphrina flavorubra* (Figure 5C). While both of the original *S. pombe* enzymes exhibit the expected activities of CC-and A-addition (Figure 5A,B), the *Taphrina* loop chimera STS added a complete CCA triplet to the tRNA substrate, as was seen with the *S. rosetta* SHS chimera, although at a rather low efficiency (Figure 5D, upper panel). Yet, on a substrate tRNA ending with CC, the loop chimera catalyzed efficient and specific addition of the terminal A (Figure 5D, lower panel), as was shown for loop chimeras of bacterial CC-adding enzymes [22] and the *S. rosetta* SHS chimera (Figure 4D). Hence, both of our loop (and B/A) insertions restored A-adding activity in the CC-adding enzymes of *S. rosetta* and *S. pombe*. To investigate whether a loop region of a more distantly related e-type enzyme is also compatible with the context of the *S. pombe* enzyme, we generated a chimeric protein carrying the flexible loop and the B/A motif of *Saccharomyces cerevisiae*. While the chimera readily incorporated two C-residues on a tRNA substrate, it completely failed to add the terminal A (Appendix A). This result corroborates our previous findings that only loop sequences of evolutionary closely related enzymes are compatible, while the corresponding region from distantly related enzymes are not exchangeable [22,24,25,33].

## 3. Discussion

### 3.1. Restricted Activities for CC- and A-Addition Are Widely Distributed in Bacteria as Well as in Eukaryotes

While bacterial tRNA nucleotidyltransferases can be divided into different functional subclasses (CCA-, CC- and A-, and CCAIIa-adding enzymes) [10,11,22,28,29,32], the existence of CC- and A-adding enzymes in a single eukaryotic species (*S. pombe*) was reported only recently [37,38]. Yet, our analysis of the genomes of choanoflagellates and *Schizosaccharomyces* indicates that separate activities for CC-and A-addition are more widespread in eukaryotes than expected (Figure 1). The independent finding of such multiple tRNA nucleotidyltransferase genes in the transcriptomes of 11 out of 21 choanoflagellates and all available genomes of *Schizosaccharomyces* sp. argues against incorrect genome assembly or annotation errors.

While the number of identified split CC-and A-adding activities in Bacteria and Eukaryotes is constantly increasing, the question remains whether there is any evolutionary benefit for an organism having enzymes with partial activities. Currently, there is no indication for such an advantage, and it is very likely that the only reason for their existence is the fact that these enzyme pairs do not provoke any evolutionary disadvantage. In other words, such combinations seem to be evolutionary neutral, according to the hypothesis coined by Kimura and others [50,51].

### 3.2. Salpingoeca rosetta Carries CC and A-Adding Enzymes of Different Evolutionary Origins

In *S. rosetta*, the two tRNA nucleotidyltransferases are of different evolutionary origins (e-type and a-type enzymes). We hypothesize that the a-type enzyme results from a horizontal gene transfer event from Alphaproteobacteria to the stem lineage of Holozoa [8] (Figure 1B,C). A direct mitochondrial origin of the a-type gene can be excluded as mitochondria became linked to Eukarya much earlier in evolution [8].

The classification into a-type and e-type enzymes is further supported by the phylogenetic analysis of our extended dataset of transcriptomes from 19 additional choanoflagellate species that splits the sequences into two groups of enzymes of different evolutionary origins (Figure 2). The fact that all Choanozoa possess an a-type enzyme supports the hypothesis that the alphaproteobacterial-like enzymes were already present in the ancestor of all Choanoflagellata. However, it is still a matter of speculation why ancestral e-type enzymes were not found in all Choanozoa (Figure 1C). One possibility is that this is the result of the loss of the corresponding genes in most metazoan groups, while it was retained in Choanoflagellata (retention scenario). Alternatively, it is possible that after being replaced by the a-type enzyme in all metazoans, the Choanoflagellata independently acquired a new e-type gene (gain scenario) [8]. However, none of the scenarios fits well with the proposed phylogeny of Choanoflagellata [39]. In addition, one has to consider that the genes have been identified in transcriptomes and are not derived from whole genome data. Hence, it cannot be ruled out that e-type genes are present in the genomes of the respective Choanoflagellata but are not constitutively transcribed.

Although at first glance, both a-type and e-type enzymes of *S. rosetta* carry all characteristic motifs for CCA-addition, these enzymes exhibit separate CC-and A-adding activities. Detailed phylogenetic analysis allowed us to pin down the evolutionary origins of this pair of enzymes. The phylogenetic distribution of *S. rosetta* CC- and A-adding enzymes excludes gene duplication events. Rather, the enzymes are the result of a horizontal gene transfer with subsequent subfunctionalization into CC-and A-adding activities. Thus, *S. rosetta* carries CC-and A-adding enzymes of different evolutionary origins (CC-adding enzyme: alphaproteobacterial origin; A-adding enzyme: ancestral eukaryotic origin). Whether all Choanoflagellata that retained both a-type and e-type enzymes share the same functional separation into A-and CC-adding enzymes cannot be inferred from the sequence alignment as all enzymes carry the B/A motif and a loop element. Nevertheless, it is very likely that other choanoflagellate e-type/a-type enzyme pairs also represent A-and CC-adding enzymes, as the a-type forms show considerable deviations of the loop consensus sequence for eukaryotic a-type CCA-adding enzymes (Figure 4C; Appendix A). Accordingly, it seems that there is a clear separation of a-type CC-adding enzymes and e-type A-adding forms. Yet, as CC- as well as A-adding enzymes of *S. pombe* represent e-types, there is obviously no mechanistic or structural cause for the observed separation in Choanoflagellata.

A similar random event in evolution seems to be the gain or loss of a- or e-type enzymes. Generally, both are highly efficient catalysts for CCA-addition, represent—in most cases (*E. coli* is an exception, as all tRNA genes encode the CCA sequence)—essential enzymes for cell viability and show comparable kinetic parameters [52,53,54]. Hence, there are no biochemical data on the superiority of a-or e-type enzymes, and the gain or loss of one type cannot be explained at the functional level.

### 3.3. CC-and A-Adding Enzymes in S. pombe Are the Result of a Recent Gene Duplication Event in the Common Ancestor of Schizosaccharomyces

Previous studies [8,35] predicted the presence of two tRNA nucleotidyltransferases in *S. pombe* and recent investigations identified these enzymes as CC-and A-adding activities [37,38]. The fact that all available genomes of the genus *Schizosaccharomyces* encode for two candidate tRNA nucleotidyltransferases, probably with CC-and A-adding function, indicates that the *S. pombe* situation is not an isolated incident. The phylogenetic network clearly separates enzymes of bacterial and ancestral eukaryotic origins irrespective of their functionality (Figure 3). This indicates that (i) subfunctionalization into CC-and A-adding enzymes in *S. pombe* is independent of the functional diversification in Bacteria and excludes horizontal transfer of a bacterial tRNA nucleotidyltransferase gene to *Schizosaccharomyces*. Rather, (ii) both *S. pombe* CC-and A-adding enzymes are of the ancestral eukaryotic type (e-type) and are very likely the result of a gene duplication event and subsequent subfunctionalization into CC-and A-adding activity, as suggested by Reid et al. [37]. Such gene and genome duplications are not uncommon in Fungi [55]. However, the closest relatives of *S. pombe*, i.e., all genome-sequenced Taphrinomycotina, encode for only one enzyme. We therefore propose that (iii) the gene duplication event occurred rather recently in the common ancestor of *Schizosaccharomyces*, as supported by the corresponding phylogenetic network, where the two e-type enzymes of *Schizosaccharomyces* sp. are located at separate subtrees. Concerning topology and branch length, these subtrees are highly similar, indicating that both enzymes evolved at the same rate (Figure 3). Furthermore, the subtree topology resembles the species phylogeny of *Schizosaccharomyces*. Hence, these data exclude a horizontal gene transfer event and indicate an early subfunctionalization after gene duplication—an event that probably occurred in all other *Schizosaccharomyces* as well.

### 3.4. The Flexible Loop—Equally Important for the Evolution of CC-Adding Enzymes in Bacteria as Well as in Eukaryotes

While the deletion of the flexible loop is a hallmark for bacterial CC-adding enzymes, this element is present in all CCA-adding enzymes. Surprisingly, it is also found in the CC-adding enzymes of *S. rosetta* and *S. pombe*. Compared to the catalytic core motifs A to E, the loop sequence is less conserved, but splits into sequence families that can be assigned to Vertebrates, Fungi, Plants, or individual bacterial classes [25]. Accordingly, the loop sequence of CCA-adding enzymes shows no common signature motif. Rather, the amino acid composition is highly variable between phylogenetic groups [25]. There are strong indications that the loop represents not just a hinge element, but acts as a lever, adjusting the amino acid template (EDxxR) for either CTP or ATP recognition. However, due to its low sequence conservation, this lever interaction can vary from loop family to loop family. In the human CCA-adding enzyme, experimental evidence and molecular modeling point towards a salt bridge between an arginine residue in the loop and the glutamate position of the amino acid template [25]. Another interaction is described in the *T. maritima* enzyme, where the crystal structure shows a hydrophobic interaction between a tyrosine residue of the loop and the aspartate located in the amino acid template [24]. Yet, neither of the loop positions is invariant. These differences, as well as the different loop families, demonstrate the variability of this interaction and, consequently, the current lack of a detailed understanding how loop and amino acid template communicate. Hence, for re-establishing the full catalytic activity in a CC-adding enzyme, one has to choose the loop sequence to be inserted very carefully, as exemplified by the unsuccessful loop chimeras between *E. coli* and *Homo sapiens*, *E. coli* and *Geobacillus stearothermophilus* [25], *Aquifex aeolicus* and *Thermotoga maritima* [24], or *S. pombe* and *S. cerevisiae* (Appendix A). As no experimentally verified CCA-adding enzymes from organisms closely related to *S. rosetta* are available, we chose the loop sequence of the human CCA-adding enzyme, as Metazoa represent the closest relatives to Choanoflagellata in the stem lineage of Holozoa [8]. The replacement of the loop region of the *S. rosetta* CC-adding enzyme by the corresponding sequence of the human enzyme resulted in a chimera with CCA-adding activity, indicating the compatibility of the chosen loop sequence (Figure 4C), as was also demonstrated for a *Bacillus* CC-adding enzyme [22]. However, this chimeric enzyme form seems to be less specific in nucleotide selection if only ATP is offered as it adds a single A residue to a tRNA substrate without a 3’-CC terminus (Figure 4D). A further extension of such a tRNA-A is not possible as the B/A motif cannot position a tRNA primer ending with a base other than C for further nucleotide additions [22,27]. Similar in vitro misincorporations are also described for the wild type CCA-adding enzyme of *E. coli* [23,41,42].

As indicated above, the identified eukaryotic CC-adding enzymes are not characterized by a loop deletion. In the a-type *S. rosetta* CC-adding enzyme, several dramatic changes in the putative Holozoa loop consensus sequence “DGRbAxV” (b is a basic and x is any amino acid; Figure 4A,C) in the N-terminal part of this region seem to be responsible for its restricted activity. Yet, it is not surprising that the chimeric enzyme carrying the human loop sequence (including the consensus) shows no complete turnover on a tRNA substrate compared to a wild type CCA-adding enzyme. Rather, it indicates that the human loop is not optimally adapted to the context of the *S. rosetta* enzyme and exhibits only limited interaction with the amino acid template and/or its surrounding region [24,25].

In contrast to the choanoflagellate *S. rosetta*, the CC-adding enzyme of the fungus *S. pombe* is not of bacterial but of ancestral eukaryotic origin. Nevertheless, this e-type CC-adding enzyme is also characterized by major deviations in the flexible loop consensus sequence for fungal CCA-adding enzymes, including the B/A motif located immediately upstream of the loop (Figure 5A). This B/A motif is involved in proper positioning of the tRNA 3′ end for terminal A-addition, forming hydrogen bonds to the 3’-CC terminus of the tRNA [25,27]. However, the restoration of this motif alone (RHH to RHE) is not sufficient for establishing an A-adding activity in this enzyme, indicating that additional sequence deviations restrict its activity to CC incorporation [37].

Compared to the loop of a-type CCA-adding enzymes (with the consensus sequence “DGRbAxV”), flexible loops of e-type CCA-adding enzymes from Fungi are characterized by the consensus sequence “Yxxx(x)SRxP” (x is any amino acid) [25]. Similar to the situation in *S. rosetta*, the flexible loop of the *S. pombe* CC-adding enzyme deviates from the consensus sequence. Hence, we assumed that changes in both loop and B/A motif are the reason for the missing A-adding activity. Accordingly, the replacement of the B/A and the loop sequence with the corresponding regions of the putative *T. flavorubra* CCA-adding enzyme generated a chimera that catalyzed the incorporation of the terminal A residue. By contrast, the insertion of the *S. cerevisiae* loop (including the B/A motif) into the *S. pombe* CC-adding enzyme did not result in the restoration of A-addition (Appendix A). While these organisms all belong to the large group of Ascomycota, *S. pombe* and *S. cerevisiae* diverged approximately 350 million years ago [56]. *T. flavorubra*, however, belongs to the closest relatives of *Schizosaccharomyces* [57]. As a result, the evolutionary relation of *S. pombe* and *S. cerevisiae* might be not close enough for functional loop compatibility.

In both *S. rosetta* as well as *S. pombe* enzyme chimeras, the A-addition is less efficient compared to original CCA-adding enzymes, as was also observed for other loop chimeras between enzymes of *B. halodurans*/*B. subtilis*, *E. coli*/*Wigglesworthia glossinida* and *H. sapiens*/*Drosophila melanogaster* [22,56]. Hence, it is possible that other enzyme parts important for A-addition are also affected in the CC-adding enzymes. Besides the B/A motif and the flexible loop, a springy hinge region consisting of interacting α-helical elements located in motif C helps to define the number of nucleotides to be incorporated and contributes to the structural rearrangements of the NTP binding pocket, although no direct interaction with the loop region could be demonstrated [24,33,58]. As the sequence variation of this motif between CC-and CCA-adding enzymes is identical to that found within the CCA-adding enzymes, it is highly unlikely that motif C mutations are responsible for the restricted activity of the investigated CC-adding enzymes. This is supported by the fact that loop insertions alone can restore A-addition, while motif C remained unchanged in these chimeras. A further explanation for the lower efficiency in A-addition is that the inserted loop region is not fully adapted to the enzyme context to adjust the NTP binding pocket correctly as it could not co-evolve to adapt full compatibility.

Accordingly, several elements in the CC-adding enzymes of both Bacteria and Eukaryotes contribute to restricted activity. The B/A motif is required for positioning the 3′ CC end of the tRNA primer for a nucleophilic attack on the ATP to be incorporated [22,25,27]. Yet, this element alone is not sufficient, as demonstrated by Reid et al. in the case of *S. pombe* CC-adding enzyme [37]. The flexible loop is essential to induce the rearrangement of the nucleotide binding pocket, where the amino acid template EDxxR is adjusted to interact with ATP instead of CTP, and its deletion converts a CCA-adding enzyme into a CC-adding enzyme [22,23]. To this end, the loop probably interacts with regions in or close to the nucleotide binding pocket [24,25]. However, there are manifold different solutions for such an interaction, as only enzymes from closely related species carry compatible loop sequences—a strong indication for co-evolution of loop and interacting enzyme regions [25]. Due to its flexibility, the loop region is disordered and not resolved in most of the crystal structures of class II CCA-adding enzymes. Based on structural models of this region, it was suggested that the deviating loop sequence in the *S. pombe* CC-adding enzyme forms a short β-sheet that restricts its flexibility and, as a consequence, inhibits A-addition [33,37]. In the crystal structure of the *T. maritima* enzyme, however, the loop structure is resolved and also folds into the short β-sheet [24]. As this protein represents a bona fide CCA-adding enzyme, the formation of the β-sheet structure per se cannot be responsible for a loss in A-addition. Hence, it is still a matter of debate why loop sequences evolved into individual and incompatible sequence families that are functional only in the context of an enzyme from the corresponding phylum. It is conceivable that only within sequence families, certain interactions with the amino acid template or its surroundings can be formed to induce the structural rearrangement required for A-addition [22,23,25]. Additional structural data with resolved loop elements are required to clarify this question.

Taken together, our data clearly show that the molecular basis for the restricted activity of eukaryotic CC-adding enzymes is located in two of three regions essential for A-addition. While the springy hinge in motif C still is compatible with A-incorporation, both *S. rosetta* and *S. pombe* enzymes show an accumulation of mutations in the flexible loop region. Additionally, the *S. pombe* enzyme carries a mutation in the third element, the B/A motif. The enzymes regain A-adding activity only if both of these regions are restored. Interestingly, this loop is not mutated, but deleted in the corresponding bacterial enzymes, indicating that independent evolutionary events shaped the CC-adding enzymes of bacteria and eukaryotes. Yet, in both domains of life, the transplantation of carefully chosen loop sequences can restore the catalytic function of the original CCA-adding enzymes. Future experiments will show whether A-adding enzymes, also significantly restricted in their catalytic properties, originated similarly through mutations in identical regions of bacterial and eukaryotic counterparts.

## 4. Material and Methods

### 4.1. mRNA Extraction (Salpingoeca rosetta) and cDNA Source (Schizosaccharomyces pombe)

Cells of *S. rosetta* Px1 (wild type) were kindly provided by Nicole King (University California, Berkeley, CA, USA) in liquid culture. Cells were harvested by centrifugation and total RNA was isolated using TRIzol^®^Reagent (Thermo Fisher Scientific, Waltham, MA, USA). The mRNA was isolated using the Dynabeads™ mRNA Purification Kit (Thermo Fisher Scientific). *S. pombe* cDNA was kindly provided by Christian Hammann (Jacobs University, Hamburg, Germany).

### 4.2. Cloning, Overexpression, and Purification of Recombinant tRNA Nucleotidyltransferases

The mRNA (250 ng) of *S. rosetta* was used for cDNA synthesis by SMART^®^ MMLV reverse transcriptase (Clontech, Takara Bio, Kusatsu, Japan) with 100 pmol of an oligo(dT) primer. The coding regions of *S. rosetta* and *S. pombe* tRNA nucleotidyltransferases were amplified from cDNA with gene specific primers. Products were cloned into pET28a(+). Flexible loop replacements were performed according to Just et al. [23].

Recombinant enzymes were expressed with a C-terminal hexahistidine tag in *E. coli* BL21 (DE3) cca::cam cells lacking the endogenous CCA-adding enzyme and purified as described [59].

### 4.3. Preparation of tRNA Substrates

Radioactively labeled yeast tRNA^Phe^ lacking the CCA terminus (tRNA) or ending with two C residues (tRNA-CC) was prepared by in vitro transcription in the presence of α^32^P-ATP (3000 Ci/mmol; Hartmann Analytic, Braunschweig, Germany) and purified as described [60,61].

### 4.4. In Vitro Nucleotide Incorporation Assay

In a total volume of 20 µL, 5 pmol of ^32^P-labeled tRNA^Phe^ or tRNA^Phe^-CC were incubated at 20 °C in 30 mM HEPES/KOH (pH 7.6), 30 mM KCl, and 6 mM MgCl_2_ for 30 to 120 min with up to 90 ng of recombinant enzyme in the presence of all four NTPs (1 mM each) or ATP (0.1 mM) and CTP (0.1 mM) separately, similar to assay conditions frequently described in the literature [22,24,25,62]. Reaction products were ethanol precipitated, separated on a 10% denaturing PAGE and visualized on a PhosphorImager (GE Healthcare).

### 4.5. Identification of Candidate tRNA Nucleotidyltransferases from Choanoflagellate Transcriptomes

We located coding sequences for candidate tRNA nucleotidyltransferases in the transcriptomes of choanoflagellates reported by Richter et al. [39]. Coding sequences for *M. brevicollis* as well as *S. rosetta* a-type and e-type enzymes were taken from Betat et al. [8]. Protein sequences from both *S. rosetta* a-type and e-type enzymes (euhChoSalros_a and euhChoSalros_e; Appendix A) were run against the choanoflagellate transcriptomes using tblastn. In each of the 19 investigated species, we obtained a candidate sequence for an alphaproteobacterial-like tRNA nucleotidyltransferase (a-type). Nine of the species carried an additional candidate sequence for an ancestral eukaryotic tRNA nucleotidyltransferase (e-type). The sources of sequences are specified in Appendix A with accession numbers provided where available.

### 4.6. Identification of Candidate tRNA Nucleotidyltransferases from Taphrinomycotina

Using tblastn, we searched genome sequences from Schizosaccharomycetes and their closest relatives, i.e., all other Taphrinomycotina using *S. pombe* sequences as a query. Hits that overlap with regions of annotated genes defined the selection of the annotated protein sequences. Hits that did not overlap with annotated genes were processed using ProSplign and protein sequences from close relatives as query. A list of accession numbers of the corresponding protein sequences is presented in Appendix A.

### 4.7. Alignments and Phylogenetic Networks

For analysis of the Choanoflagellata and Taphrinomycotina tRNA nucleotidyltransferase protein sequences, we added putative and/or experimentally verified sequences of a-type enzymes from the closely related groups (see Appendix A). Alignments were computed using Clustal Omega (Sievers et al. 2011 [63]) followed by construction of the phylogenetic networks using SplitsTree version 4.6 [64] with standard parameter settings.

### 4.8. Species Trees Topology

The topology of the phylogeny of Choanoflagellata shown in Figure 1C is taken from Richter et al. [39] and built on a consensus of previously published phylogenies [65,66,67]. The larger context, i.e., the relationship of choanoflagellates to Metazoa and other choanozoan groups, is adopted from Shalchian-Tabrizi [68]. The topology of the phylogeny of Taphrinomycotina (Figure 1D), is in agreement with Spatafora et al. [57].

### 4.9. Accession Numbers

Accession numbers of investigated tRNA nucleotidyltransferase sequences as well as newly identified sequences that are not annotated are presented in the Appendix A.

## Figures and Tables

**Figure 1 ijms-21-00462-f001:**
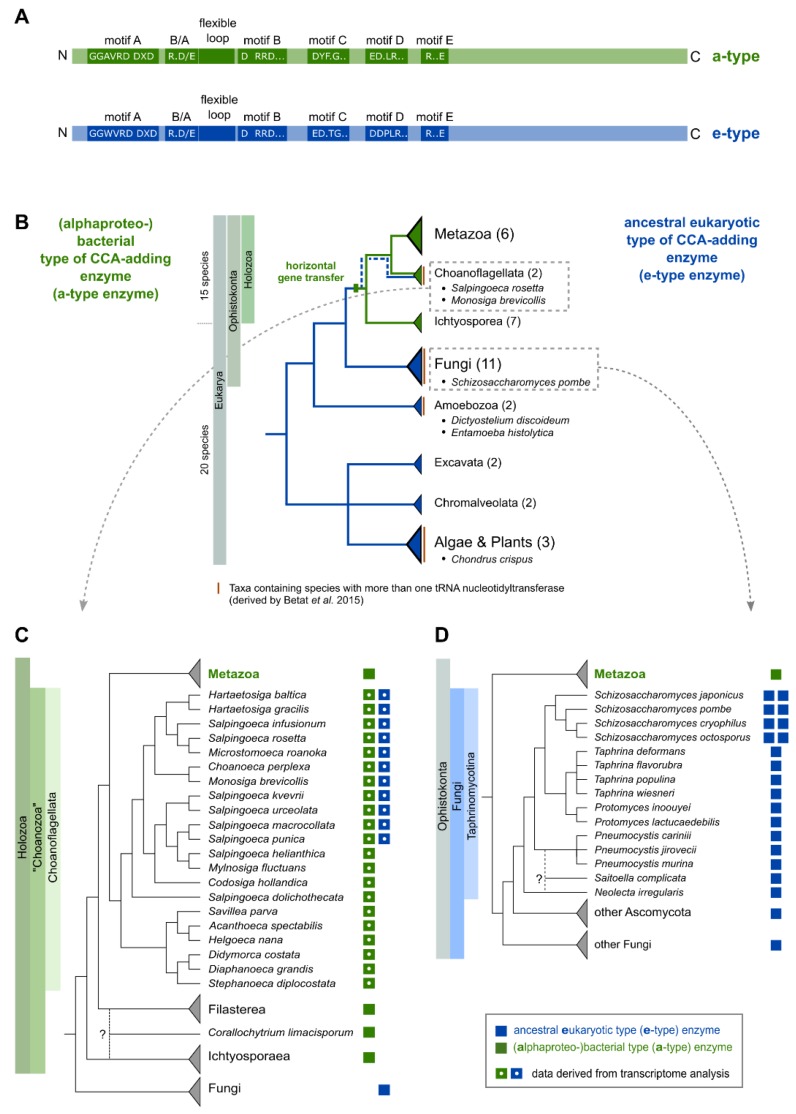
Phylogenetic distribution of eukaryotic species bearing genomes with more than one tRNA nucleotidyltransferase gene. (**A**) Schematic representation of class II tRNA nucleotidyltransferase organization. Motifs A to E of the catalytic core are indicated in green and blue. Additional motifs involved in A-addition are the basic/acidic motif (B/A; indicated as R (arginine) and D (aspartate), other basic/acidic combinations are also found) and the flexible loop, both located between motifs A and B. (**B**) Phylogenetic tree summarizing the data from 35 eukaryotic genomes analyzed in Betat et al. [8]. The tree topology represents all major eukaryotic clades. The total number of genomes analyzed per clade is given in brackets. Species names are given for genomes containing more than one candidate tRNA nucleotidyltransferase sequence. (**C**,**D**) Phylogenetic distribution of tRNA nucleotidyltransferases in Choanoflagellata and Taphrinomycotina, respectively. Boxes next to the species name indicate the type of candidate tRNA nucleotidyltransferase genes (green—alphaproteobacterial a-type; blue—ancestral eukaryotic e-type). Closed and open boxes indicate whether candidates are derived from genome or transcriptome data. The trees to the left of the species names are for illustrative purposes as the species phylogeny is not well established in some parts. Multifurcations, question marks, and dotted branches indicate uncertainty about the phylogenetic position.

**Figure 2 ijms-21-00462-f002:**
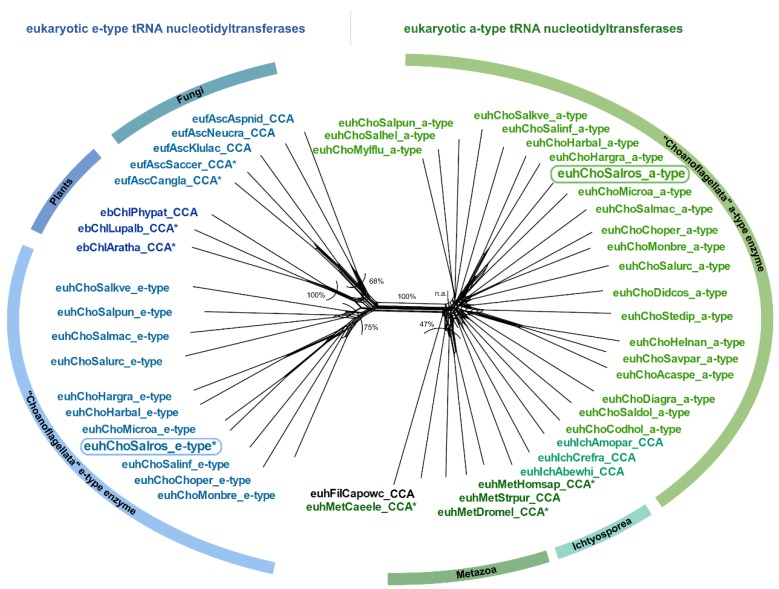
Phylogenetic network of a- and e-type tRNA nucleotidyltransferase sequences from Choanoflagellata. On the right of the main split, a-type sequences of CCA-adding enzymes from Metazoa cluster with a-type sequences from Choanoflagellata (CC-adding enzyme in *S. rosetta*). On the left of the main split, e-type enzyme sequences from Fungi and Plants cluster with candidate e-type enzyme sequences of Choanoflagellata (A-adding enzyme in *S. rosetta*). An asterisk (*) marks sequences with experimentally verified function. Frames indicate the enzymes of *S. rosetta* that were analyzed in this study. Bootstrap values are indicated as percentage values, n.a. represents no shared split. Sequence names consist of the taxonomic group prefix (eu—Eukaryota, Unikonta; euh—Eukaryota, Holozoa; eb—Eukaryota, Bikonta; euf—Eukaryota, Fungi) as well as the first three letters of the genus and species name (see Appendix A).

**Figure 3 ijms-21-00462-f003:**
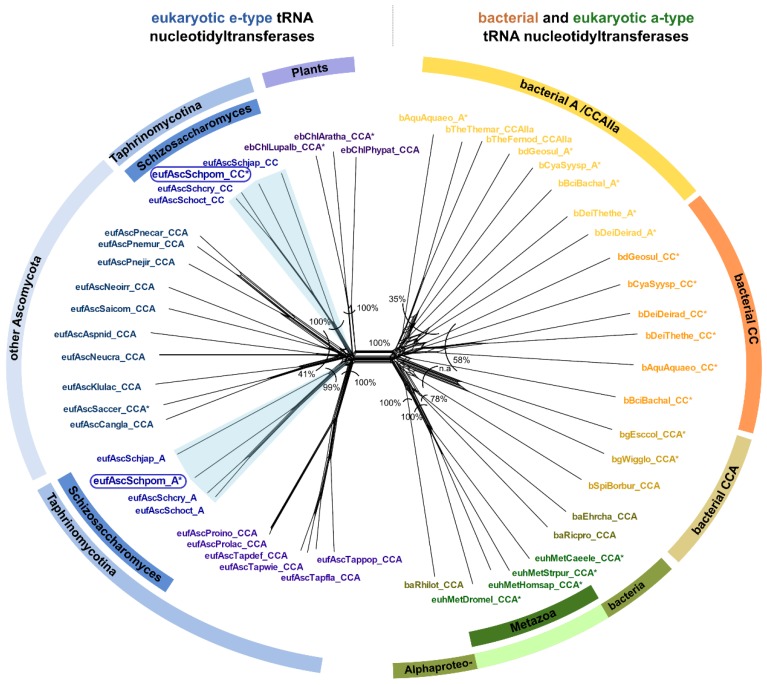
Phylogenetic network of tRNA nucleotidyltransferase sequences from Taphrinomycotina in a greater context. The split in the center separates e-type (left) and a-type (right) sequences. The left side includes all sequences from genome-sequenced Taphrinomycotina. *Schizosaccharomyces* species are represented by both confirmed and putative CC-adding and A-adding enzyme sequences. Each of the corresponding subtrees of these *Schizosaccharomyces* enzymes resembles the species phylogeny in topology and branch length, indicating that both enzymes evolved at the same rate. These data exclude a horizontal gene transfer event and indicate subfunctionalization shortly after gene duplication. The cluster on the right side includes CCA-, CC-, and A-adding enzyme sequences of bacterial origin (including metazoan a-type sequences of alphaproteobacterial origin). An asterisk (*) marks sequences that have been experimentally verified for their function. The frames indicate the investigated *S. pombe* enzymes. Bootstrap values are given as percentage values, n.a. indicates no shared split. Sequence names consist of the taxonomic group prefix (eu—Eukaryota, Unikonta; euh—Eukaryota, Holozoa; eb—Eukaryota, Bikonta; euf—Eukaryota, Fungi) as well as the first three letters of the genus and species name (see Appendix A).

**Figure 4 ijms-21-00462-f004:**
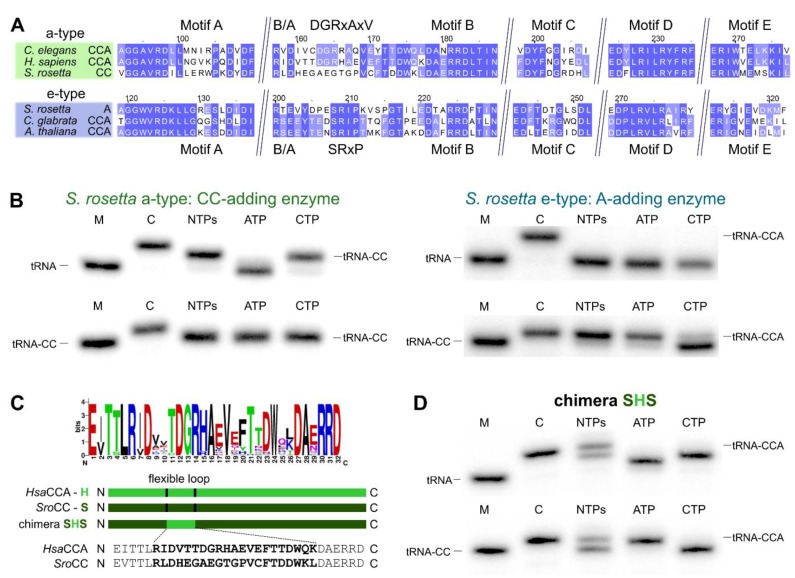
The *S. rosetta* enzymes are tRNA nucleotidyltransferases with complementing partial activities. (**A**) Alignment of *S. rosetta* a-type CC- and e-type A-adding enzymes with sequences of experimentally verified CCA-adding enzymes of *C. elegans* (a-type; accession number (AC): Q93795; [43]), *H. sapiens* (a-type; AC: Q96Q11; [44,45]), *C. glabrata* (e-type; AC: XP_449283.1; [46]) and *A. thaliana* (e-type; AC: Q94K06; [47]). The alignment is restricted to the N-terminal catalytic core elements. Highly conserved positions are indicated in blue. The alignment shows the high similarity of a-type CCA- and CC-adding enzymes as well as e-type CCA- and A-adding enzymes. The *S. rosetta* CC-adding enzyme shows strong deviations from the conserved flexible loop sequence of related a-type CCA-adding enzymes. (**B**) Radiolabeled tRNA^Phe^ (top) or tRNA^Phe^-CC (bottom) was incubated with either *S. rosetta* a-type enzyme (left) or e-type enzyme (right). Nucleotide addition on radiolabeled tRNA^Phe^ (top) or tRNA^Phe^-CC (bottom) identifies *S. rosetta* a-type and e-type enzymes as bona fide CC-and A-adding enzymes, respectively. In the presence of CTP only, the A-adding enzyme catalyzes a slight misincorporation of an additional C residue, a side reaction also observed for other tRNA nucleotidyltransferases [41,42]. (**C**) The flexible loop of *S. rosetta* CC-adding enzyme exhibits strong deviations from the consensus sequence “DGRxAxV” found in eukaryotic a-type CCA-adding enzymes. In the chimeric enzyme SHS, the corresponding region of *H. sapiens* CCA-adding enzyme (H) was transplanted into the CC-adding enzyme of *S. rosetta* (S). (**D**) Chimera SHS incorporates three nucleotides into a tRNA without CCA end and the terminal A residue into a tRNA carrying two C residues (tRNA-CC). M, mock incubation of the tRNA in the absence of enzymes; C, control tRNA^Phe^ with CCA end as size marker.

**Figure 5 ijms-21-00462-f005:**
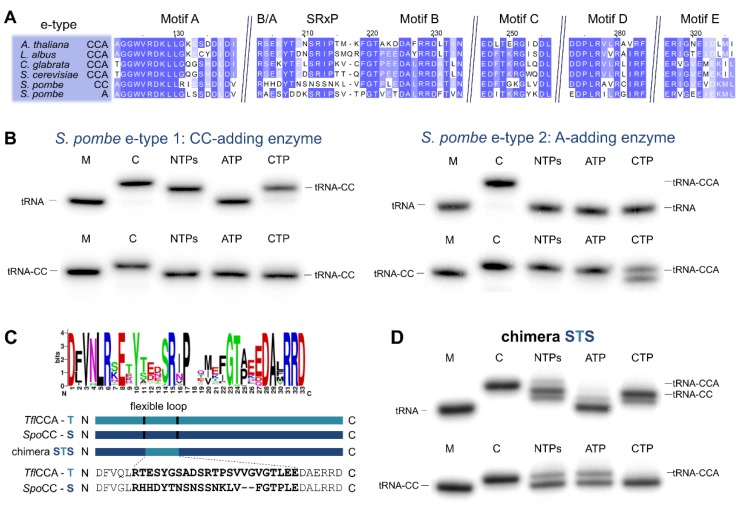
*S. pombe* performs CCA-addition with collaborating CC-and A-adding enzymes. (**A**) Alignment of the catalytic core of both *S. pombe* e-type enzymes with sequences of tested e-type CCA-adding enzymes of *A. thaliana* (accession number (AC): Q94K06; [47]), *L. albus* (AC: AAB03077.1; [48]), *C. glabrata* (AC: XP_449283.1; [46]), and *S. cerevisiae* (AC: P21269; [49]). Highly conserved residues are indicated in blue. Similar to the *S. rosetta* counterpart, the *S. pombe* CC-adding enzyme shows strong deviations from the conserved flexible loop sequence responsible for its functional restriction. (**B**) Both enzymes were incubated with labeled substrates tRNA^Phe^ (top) or tRNA^Phe^-CC (bottom). The e-type enzyme 1 (left) exclusively adds two C residues on tRNA^Phe^ but does not accept tRNA^Phe^-CC for A-addition and therefore represents a CC-adding enzyme. The e-type enzyme 2, on the other hand, accepts only tRNA^Phe^-CC as a substrate and adds just a terminal A residue and is therefore identified as an A-adding enzyme. Similar to the *S. rosetta* enzyme (Figure 4B), this enzyme misincorporates an additional C residue if CTP only is offered. Taken together, these enzymes represent typical CC-and A-adding enzymes. (**C**) In the CC-adding enzyme, the flexible loop is lacking the conserved sequence “Yxxx(x)SRxP” found in eukaryotic e-type CCA-adding enzymes. For construction of an enzyme chimera carrying the flexible loop and the B/A motif of a closely related fungal CCA-adding enzyme, the corresponding sequence of *T. flavorubra* (T) was inserted into the CC-adding enzyme of *S. pombe* (S), generating the chimera STS (S, *S. pombe* CC-adding enzyme N-terminus; T, flexible loop of *T. flavorubra* CCA-adding enzyme; S, *S. pombe* CC-adding enzyme C-terminus). (**D**) On tRNA^Phe^ lacking the 3’-CCA terminus, chimera STS incorporates three nucleotides. While the addition of the terminal residue is not very efficient, one has to consider that the inserted regions in a chimeric protein are not perfectly adjusted to the context of the host protein. Hence, wild type-like activity cannot be expected. On tRNA^Phe^-CC, the chimera efficiently adds a terminal A residue, clearly demonstrating that A-addition is restored to a considerable extent. M, mock incubation of the tRNA in the absence of enzymes; C, control tRNA^Phe^ with CCA end as size marker.

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
