# Peer review of "Divergent Evolution of Eukaryotic CC- and A-Adding Enzymes"

_ijms, 2020, doi:10.3390/ijms21020462_

Round 1

Reviewer 1 Report

The paper by Erber et al. analyzes the phylogeny and molecular basis for the existence of split CC/A adding enzymes in Eukaryotes. Quite recently, such situation was identified in the yeast S. pombe and the current study identifies another interesting example that differs from the situation in S. pombe. While in S. pombe both CC- and A adding enzymes belong to the ancestral eukaryotic (a-) type, two CC/A adding enzymes belonging to phylogenetically different types were identified in Choanoflagellata (a-type and e-type). The in vitro activities of the CC/A adding enzymes from one such species, Salpingoeca rosetta were characterized in the present study proving the CC- and A-adding enzymatic activities. The a-type was identified as CC-adding enzyme whereas the e-type represents the A-adding enzyme. Sequence comparisons allowed for the identification of a specific loop region in the CC adding enzyme that prevents it from A-addition. Modified enzymes were generated based on the CC adding enzymes from S. rosetta and S. pombe in which this loop region was replaced by that from related CCA adding enzymes and indeed full CCA addition was restored. This study addresses an interesting problem in tRNA biology and significantly extends the knowledge about the evolution of collaborating CCA adding activities. Experiments and presentation of data are convincing.

Comments to consider for further improvement

The authors identify a distinct evolutionary origin for the split CC/A enzyme situation in S. pombe and S. rosetta. While in S. pombe a gene duplication event is assumed that led to the formation of to functionally divergent CC/A adding enzymes, in Choanoflagellates a horizontal gene transfer event is assumed. Can the authors speculate about the evolutionary benefit of such events? Why was there an evolutionary pressure to select for mutations inactivating the A-incorporation after gene duplication or acquisition of the second CCA adding enzyme?

In Fig4B, mock treated tRNA-CC appears to migrate a bit faster than tRNA-CC incubated with NTP/ATP/CTP. Is this a gel running artefact?

The authors have convincingly demonstrated that the S. rosetta a-type and e-type enzymes carry CC- and A-adding activities, respectively. Have the authors analyzed whether the two enzymes together can convert tRNA with no addition to CCA? Since all the tools are established, it would be nice to demonstrate their collaboration in CCA addition.

Author Response

Reviewer 1:

The authors identify a distinct evolutionary origin for the split CC/A enzyme situation in pombe and S. rosetta. While in S. pombe a gene duplication event is assumed that led to the formation of two functionally divergent CC/A adding enzymes, in Choanoflagellates a horizontal gene transfer event is assumed. Can the authors speculate about the evolutionary benefit of such events? Why was there an evolutionary pressure to select for mutations inactivating the A-incorporation after gene duplication or acquisition of the second CCA adding enzyme?

      Response:

      It is difficult to speculate about an evolutionary benefit of having two separate activities for CC- and A-addition. It is very likely that such a system evolved or was maintained simple due to the fact that there is obviously no disadvantage of a two-enzyme system – a situation that is described by the neutral theory of evolution, according to M. Kimura. We have added such a statement in the Discussion section 3.1 (page 10, lines 304-309).

In Fig4B, mock treated tRNA-CC appears to migrate a bit faster than tRNA-CC incubated with NTP/ATP/CTP. Is this a gel running artefact?

      Response:

      The reviewer is right – this migration difference is only a gel running artefact. If one extrapolates from the right corner for the mock band to the band in the ATP lane, one clearly sees that both bands migrate at identical positions.

The authors have convincingly demonstrated that the rosetta a-type and e-type enzymes carry CC- and A-adding activities, respectively. Have the authors analyzed whether the two enzymes together can convert tRNA with no addition to CCA? Since all the tools are established, it would be nice to demonstrate their collaboration in CCA addition.

      Response: We thank the reviewer for this great idea. We have performed the corresponding experiment and show now in Figure S1 that the CC-and A-adding enzymes of. S. rosetta nicely collaborate in vitro, resulting in the synthesis of a complete CCA-sequence (Results, page 7, lines196-197).

Reviewer 2 Report

The authors present a phylogenetic analysis leading to the finding that the choanoflagellate Salpingoeca rosetta carries a CC- and an A-adding enzyme, the second eukaryote with this feature. Further analysis reveals that the two enzymes are derived from evolutionary distinct progenitors and can be assigned to the e- or a-type respectively. Introduction of a flexible loop found in the human CCA-adding enzyme restores CCA-adding activity to the otherwise CC-adding S. rosetta protein. The same result could be found for Schizosaccharomyces pombe CC-adding enzyme after more careful selection of an appropriate loop donor.

This is manuscript highlights the similarities in protein architecture and mechanism, by showing that transplantation of a loop region is sufficient to engineer extended activities. The independent acquisition of split CC- and A- adding activity in two eukaryotes is quite interesting and convincingly described. While experiments are kept simple and on point, the introduction and discussion deliver sufficient context and explanations to make it a convincing story. I would appreciate if the following points could be addressed:

1) Figure 4D: the chimera seems to lose specificity the presence of ATP leads to a modification of the tRNA end without CC. Two bands are visible upon incubation with the dNTPs mix despite levels of ATP being higher in the dNTP mix than in ATP incubation condition alone. Can these observations be explained by the suggested altered protein architecture?

2) Are there any indications whether it would be beneficial for choanoflagellates to split CC and A-adding activity? Do sequences motives point towards different cellular localizations or is tRNA quality control involved?

3) I do not completely understand the expression system. The authors used E. coli lacking endogenous CCA-adding enzyme to express S. rosetta proteins but expression of CC-adding enzyme alone should not rescue the expression host.

4) Is it possible with the results from this study to predict from sequences alone whether other Choanoflagellata carrying e- and a-type are likely having the same distribution of functionality as found here for S. rosetta? Is it possible that CC- and A-adding activities of e- and a-type are switched in other species?

5) Is there any biochemical data on the superiority of a-type enzymes, for example in regard of catalytic efficiency or stability, explaining the loss of e-type enzyme?

Author Response

Reviewer 2:

Figure 4D: the chimera seems to lose specificity the presence of ATP leads to a modification of the tRNA end without CC. Two bands are visible upon incubation with the dNTPs mix despite levels of ATP being higher in the dNTP mix than in ATP incubation condition alone. Can these observations be explained by the suggested altered protein architecture?

      Response: As it was observed for wild type tRNA nucleotidyltransferases as well, misincorporations in in vitro assays can occur, especially in the presence of only one type of NTPs. However, the chimera is only able to add a single A residue, as the B/A motif cannot position a tRNA primer ending in a base other than C for further nucleotide additions. Hence, the observed single A-addition on a tRNA lacking the CC is a clear in vitro artifact known from other CCA-adding enzymes was well. We have now included this point in the Results section (page 8, lines 239-242) as well as in the Discussion (page 12, lines 403-408).

Are there any indications whether it would be beneficial for choanoflagellates to split CC and A-adding activity??

      Response: This highly interesting question is identical to question 1 of reviewer 1. We have added this topic and the neutral theory of evolution in the Discussion 3.1 (page 10, lines 304-309).

Do sequences motives point towards different cellular localizations or is tRNA quality control involved?

Response: A Mitoprot-based prediction indicates that the choanoflagellate enzymes carry N-terminal mitochondrial target sequences. However, the accuracy of such computer-based predictions is only moderate and the calculated import probability varies from enzyme to enzyme, so that such results are not very reliable. Yet, the only available mitochondrial genome of a choanoflagellate (M. brevicollis) carries only tRNA genes that lack a CCA-end. Hence, the CCA-adding activity (of a single enzyme or a pair of CC- and A-adding enzymes) has to be imported to ensure mitochondrial translation. We have included this statement in the Results section (page 6-7, lines 175-184.

I do not completely understand the expression system. The authors used E. coli lacking endogenous CCA-adding enzyme to express S. rosetta proteins but expression of CC-adding enzyme alone should not rescue the expression host.

      Response: As all tRNA genes of E. coli already encode the CCA end, the CCA-adding enzyme is not essential for viability. It is just required for CCA-end repair and tRNA quality control. We have generated several E. coli expression strains that lack an endogenous CCA-adding enzyme, so that we can be sure that the activity of the recombinantly expressed and purified enzyme forms is not an artifact due to a contaminating endogenous CCA-adding enzyme.

Is it possible with the results from this study to predict from sequences alone whether other Choanoflagellata carrying e- and a-type are likely having the same distribution of functionality as found here for rosetta? Is it possible that CC- and A-adding activities of e- and a-type are switched in other species?

      Response: Yes, it is very likely that the other choanoflagellate enzyme pairs also carry split activities, as we find considerable loop sequence deviations in the a-type enzymes, indicating a restricted activity of CC-addition, as we identified it in S. rosetta. Furthermore, it is probably an evolutionary accident that the a-type enzymes of a choanoflagellate ancestor evolved into CC-adding enzymes, as there is no indication that e- or a-type enzymes differ significantly in their catalytic properties. Rather, their kinetic parameters are quite similar, and it is possible that in unidentified choanoflagellate species, an e-type enzyme evolved into a CC-adding activity, as it occurred in S. pombe. We have included this statement in the Discussion section (page 10, lines 339-345) and loop sequences of arbitrarily chosen choanoflagellate species in the supplementary.

Is there any biochemical data on the superiority of a-type enzymes, for example in regard of catalytic efficiency or stability, explaining the loss of e-type enzyme?

Response: See our answer above – the kinetic parameters of e- and a-type CCA-adding enzymes are in a very similar range, indicating that both enzymes are highly active and show no differences in functionality of catalytic efficiency. Hence, gain or loss of one type of enzymes cannot be explained by biochemical or catalytic differences of these two enzyme forms. We have included this in the Discussion (page 11, lines 346-350).